# A Relationship between Micro-Meteorological and Personal Variables of Outdoor Thermal Comfort: A Case Study in Kitakyushu, Japan

**Dadang Hartabela [1,2,\*]**, **Bart Julien Dewancker [3]** and **Mochamad Donny Koerniawan [4]**

1   Graduate School of Environmental Engineering, The University of Kitakyushu, Kitakyushu 808-0135, Japan
2   Department of Architecture, University of Bandar Lampung, Bandar Lampung 35142, Indonesia
3   Department of Architecture, The University of Kitakyushu, Kitakyushu 808-0135, Japan; bart@kitakyu-u.ac.jp
4   Department of Architecture, Bandung Institute of Technology, School of Architecture, Planning and Policy Development, Bandung 40132, Indonesia; donny@ar.itb.ac.id
\*   Correspondence: dadanghartabela@gmail.com; Tel.: +81-70-7620-2157

**Abstract:** Outdoor thermal comfort is an important indicator to create a quality and livable environment. This study examines a relationship between micro-meteorological and personal variables of outdoor thermal comfort conditions in an urban park. The data collection of outdoor thermal comfort is carried out using two methods in combination: micro-meteorological measurement and questionnaire survey. This finding shows that most of the respondents were comfortable with the thermal, wind, and humidity condition. The acceptability and satisfaction level of thermal comfort were positive. The most significant micro-meteorological variable for the physiologically equivalent temperature (PET) value is mean radiant temperature (Tmrt). As the Tmrt value is influenced by how much shading is produced from the presence of vegetation or buildings around the measurement location, this finding shows that the shadow was very important to the thermal comfort conditions in the Green Park Kitakyushu. The most influential micro-meteorological variable for the three different personal variables (TSV, WFSV, and HSV) is air temperature. The strongest relationship among the four variables is between TSV and PET. The findings will be the basis for the city authorities in preparing regional development plans, especially those related to the planning of city parks or tourist attractions.

**Keywords:** outdoor thermal comfort; environment factor; human factor; thermal perception; thermal sensation; thermal comfort; urban park; outdoor urban space

## 1. Introduction

### 1.1. Urban Parks and Urban Heat Island Phenomenon

Half of the world's population lives in cities [1], this demands a quality and livable environment inside the city. Cities occupy 2% of the earth's surface but their inhabitants consume 75% of the world's energy resources [2]. Some cities experience problems with the thermal quality of their environment. Kolokotroni [3] stated, in the city of London there was an increase in temperature due to the Urban Heat Island (UHI) phenomenon, the cooling load in the city was 25% higher than in rural environments, whereas the heating load diminished by 22%. The same phenomenon is also found in other cities. In the U.S., on a yearly average, urban areas are found to be substantially warmer than the non-urban fringe by 2.9 °C, except for urban areas in biomes with arid and semiarid climates [4]. Moreover, the average UHI amplitude is remarkably asymmetric with a 4.3 °C temperature difference in summer and only 1.3 °C in winter [4]. The UHI phenomenon is generally seen as being caused by an increase in sensible heat in urban areas as vegetated and evaporating soil surfaces are replaced by relatively impervious low albedo paving and building materials and a reduction in latent heat flux [4]. So that, an outdoor thermal comfort evaluation of an urban area is important to mitigate the increasing of UHI.

### 1.2. Thermal Comfort Studies

The study of thermal comfort began in the mid-1930s when Winslow, Herrington and Gagge laid the foundations for human thermoregulation and partition calorimetry [5]. The American Society of Heating, Refrigerating and Air Conditioning Engineers (ASHRAE) defines thermal comfort as a state of mind that expresses satisfaction with the thermal environment [6]. This definition provides a physiological and sensory basis for the concept of "thermal comfort" [7]. In indoor environment, the range of thermal comfort for neutral temperature sensation is between 28 °C and 30 °C, where there is an absence of temperature regulatory effort by sweating, vasoconstriction, and vasodilation [7].

The study of outdoor thermal comfort has been carried out by many experts [8–12]. Studies on the impact of shading, the presence of trees, and vegetation on decreasing city temperatures show a positive effect during the day [13,14]. Many evaluations and simulations of the city's temperature cooling performance through vegetation have also been carried out [15–18]. Tan, Liao, Bedra, and Li [15] evaluated the 3D cooling performances of the three vegetation combination scenarios in the urban area using the ENVI-met model. Based on this study, shadows can directly affect the 3D cooling effect of the vegetation combination. The larger the shaded area, the better the cooling effect for the same vegetation cover [15]. A study in Sao Paulo found that during autumn, April 2013, the average maximum air temperature difference reached 0.5 °C and in February 2014, during the extreme warm summer, air temperature differences became more significant, and the effect of vegetation was slightly more pronounced showing maximum air temperature differences up to 0.6 °C [14].

An outdoor thermal comfort study in Taiwan [19] found that there are three categories of thermal comfort values based on the PET (physiologically equivalent temperature) value, namely: thermal suitable (PET between 22–34 °C), thermal stress (PET > 38 °C), and cold stress (PET < 18 °C). Other study found the thermal acceptable range for an entire year was 21.3–28.5 °C PET [20]. While in Qinghai-Tibet Plateau (a cold highland area) using three categories of thermal conditions that are still acceptable to the community, namely PET 13–18 °C (slightly cool), PET 18–23 °C (neutral), and PET 23–29 °C (slightly warm) [21].

Research on thermal comfort is divided into two types of environmental condition: indoor and outdoor. Several indices are being used to calculate thermal comfort, such as new effective temperature (ET*) [22], operative temperature [6], and standard effective temperature (SET*) [23], Out_SET* [6,24], Universal Thermal Climate Index (UTCI) [8,25,26], PET [27,28], and outdoor environmental heat index (OEHI) [29]. For the indoor environment, the most popular index is predicted mean vote (PMV) [30].

For outdoor research, one of popular index used in the subtropics is PET [31,32]. This is because PET is developed considering the effects of short and long wave radiation fluxes in the external environment on human heat balance [33]. PET utilized mean radiant temperature, clothing, and metabolic rates of users as input values of outdoor thermal comfort [11]. PET also enables person to compare the effects of the outdoor thermal conditions based on his/her own indoor experiences [28]. Another advantage is PET uses a commonly known degree (°C) to calculate the thermal comfort index which is suitable in various climates [34].

The PET variables conclude four environmental parameters (air temperature, humidity, wind, and mean radiant temperature) and two personal variables (clothing insulation level and metabolic rate or activity level). Earlier [35], PET did not consider clothing and activity levels as variables, but later in RayMan Model [36,37] (an outdoor thermal comfort software), these variables are added. Based on these reasons, this study uses PET as an outdoor thermal comfort index.

The outdoor thermal comfort variables are divided into two types: (1) micro-meteorological variables; and (2) personal variables (see Figure 1). As for the micro-meteorological variables has four variables: air temperature (Ta), relative humidity (RH), wind velocity or wind speed (v), and mean radiant temperature (Tmrt or MRT). The personal variables are metabolic heat (M), clothing insulation (Icl), and a questionnaire survey which consists

of respondents' thermal comfort condition during the survey (e.g., thermal sensation and acceptability) and demographic backgrounds (e.g., gender and age).

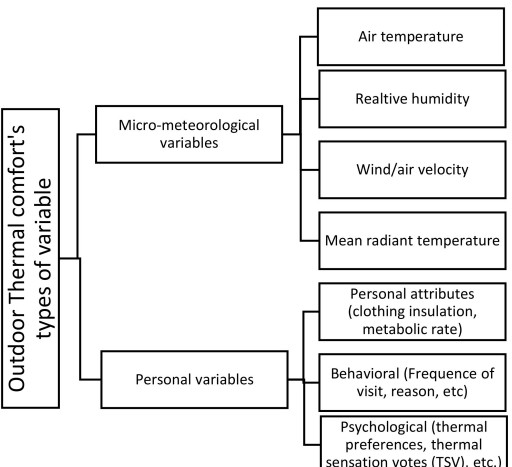

**Figure 1.** The types of the variable of outdoor thermal comfort.

There are several uses of the perception index of thermal comfort, including thermal sensation (TS-Givoni) [38], thermal sensation vote (TSV) [6], optimum thermal environment (OTE) [10], thermal perception classification (TPC) [19], and human thermal sensation (HTS) [39]. The most popular index to calculate thermal comfort perception in subtropics is TSV [31]. According to this reason, this study uses TSV as a thermal perception index.

The TSV are rated on the ASHRAE 7-point scale [40] and ISO 7730 [41]. The 7-point sensation scale ranges from "cold" (−3), "cool" (−2), "slightly cool" (−1), "neutral" (0), "slightly warm" (+1), "warm" (+2) and "hot" (+3) conditions. Other researchers [42,43] use 9-point scale by adding "very cold" and "very hot" (see Table 1). The main point of these scales is to give an optional range of answer of the actual thermal sensation that was felt by the respondents during the research. The actual thermal discomfort limit can be determined based on the user's perception. Thus, the results of this TSV survey will be compared with survey results based on measurements using thermal measuring instruments in the field. This study uses a 7-point scale of thermal sensation.

**Table 1.** Thermal sensation scale [1].

| 7-Point Scale | | 9-Point Scale | |
|---|---|---|---|
| | | Very hot | 9 |
| Hot | 3 | Hot | 8 |
| Warm | 2 | Warm | 7 |
| Slightly warm | 1 | Slightly warm | 6 |
| Neutral | 0 | Neutral | 5 |
| Slightly cool | −1 | Slightly cool | 4 |
| Cool | −2 | Cool | 3 |
| Cold | −3 | Cold | 2 |
| | | Very cold | 1 |

[1] This table is adapted from Mukherjee and Mahanta [44].

### 1.3. Objectives of the Study

This research is a development study of outdoor thermal comfort from the previous findings, especially for subtropical areas, such as the city of Kitakyushu. The findings will be the basis for the city authorities in preparing regional development plans, especially those related to the planning of city parks or tourist attractions.

This study examines a relationship between micro-meteorological variables and personal variables of outdoor thermal comfort in an urban park. The micro-meteorological variables are air temperature, (Ta), relative humidity (RH), wind velocity (v), and mean

radiant temperature (Tmrt). The personal variables are activity level/metabolic rate (M), and clothing insulation level (I*cl*). In this study, PET and thermal sensation vote (TSV) are used as the thermal comfort indices. This study also introduces new two indices, they are wind flow sensation vote (WFSV) and humidity sensation vote (HSV).

To facilitate understanding of the material, the following questions were used.

(1)  How are the people's perceptions of outdoor thermal sensation (TSV), wind flow sensation (WFSV), and humidity sensation (HSV)?
(2)  How are the acceptability and satisfaction level of outdoor thermal comfort?
(3)  How are the satisfaction preferences for shading, sunlight, and wind condition?
(4)  What is the most significant micro-meteorological variables for PET?
(5)  How is the relationship between micro-meteorological and TSV?
(6)  How is the relationship between micro-meteorological and WFSV?
(7)  How is the relationship between micro-meteorological and HSV?
(8)  How is the relationship between PET and personal variables (TSV, WFSV, and HSV)?

## 2. Materials and Methods

### 2.1. Urban Park's Description

Urban parks are physical environments where people living in the city, who have different cultures and socioeconomic status, come together in their leisure time and commune with nature; which are organized for physical, ecological, psychological, and recreational purposes; which bear active and passive outdoor activities such as meeting, entertainment, recreation etc., which help reduce the stresses of urban life [45]. The quality of urban parks is directly related to the level of realization of optional activities among the outdoor activities, which can be assessed under three headings as: necessary activities, optional, and social activities [46].

According to the Ministry of Land, Infrastructure, Transport, and Tourism (MLIT) of Japan, the city park is divided into five categories, they are basic parks for community use, basic parks for city wide use, large-scaled parks, national government parks, and buffer green belts [47]. Where the urban park is associated to the basic parks for city-wide use or the large-scaled parks. There are two types of basic parks for city wide use, a comprehensive park (for use by all residents in a city for various purposes, the standard area ranges from 10 to 50 ha according to the size of the city) and a sport park (mainly for athletic activities, from 15 to 75 ha). For the large-scaled parks, it is divided into two types: regional park and recreation cities. The regional park has the standard area at least 50 ha and its recreational facilities are placed organically, while the recreation cities are areas where a variety of recreation facilities are provided mainly for the entire population area of a large city or other city and has a total area of about 1000 ha. In 2005, there are 1973 basic parks for city-wide use and 190 large-scaled parks in all over Japan.

Based on the legal classification of Japanese parks, there are two types of park, they are natural park and urban park [47]. Urban parks are created by central government or local bodies who acquire a certain area of land and open it for public use. While natural parks remain the property of various private individuals, and its natural landscape is maintained by legislation restricting land use.

In Kitakyushu city area, there are 23 parks (see Table 2) which are mentioned on the official tourism information website of Kitakyushu City [48]. Based on the criteria of urban park, it is found that the Hibikinada Green Park Kitakyushu is the most eligible for study case. This park has fulfilled the criteria of large-scale park and has the legal classification as an urban park. It became an important criteria for mitigating the UHI as researchers suggest that the urban park should have a large area size [49,50]. An investigation of large urban park cooling effects in Madrid showed that large-scale urban parks generally play a significant part in creating a cognitive state of high-perceived thermal comfort spaces for residents [50]. While in Wroclaw it was found that the cooling distance varied from 110 m to 925 m depending on park size, forest area, and land use type in the park's vicinity [49].

**Table 2.** List of parks in Kitakyushu City area.

| Name of Park | Location (Ward) | Area Size * (ha) | Park Type ** | Legal Classification ** |
|---|---|---|---|---|
| Itozu-no-mori Zoological Park | Kokurakita | 10.82 ha | Buffer Green Belts (Specific parks for zoos) | Urban Park |
| Hibikinada Green Park | Wakamatsu | 66.91 ha | Large-scaled parks (Regional parks) | Urban Park |
| Agriculture and Livestock Information and Research Center (Hananooka Park) | Kokura-minami | 9.88 ha | Buffer Green Belts (Specific parks for agriculture) | Urban Park |
| Kawachi Wisteria Garden | Yahatahigashi | 2.17 ha | Buffer Green Belts (Specific parks for botany) | Urban and Natural Park |
| Shiranoe Botanical Gardens | Moji | 8.91 ha | Buffer Green Belts (Specific parks for botany) | Urban and Natural Park |
| Mekari Park | Moji | 48.78 ha | Basic Parks for City Wide Use (Comprehensive parks) | Urban and Natural Park |
| Hiraodai Countryside Park | Kokura-minami | 25.32 ha | Basic Parks for City Wide Use (Comprehensive parks) | Urban Park |
| Adachi Park | Kokurakita | 7.27 ha | Buffer Green Belts | Natural Park |
| Takatoyama Park | Wakamatsu | 4.86 ha | Basic Parks for City Wide Use (Comprehensive parks) | Urban and Natural Park |
| Kisshoji Park | Yahatanishi | 5.21 ha | Buffer Green Belts (Specific parks for botany and history) | Urban and Natural Park |
| Yamada Green Zone/Yamada Park | Kokurakita | 10.06 ha | Buffer Green Belts (Specific parks as a scenic park) | Natural Park |
| Tamukeyama Park | Kokurakita | 11.03 ha | Buffer Green Belts (Specific parks as a scenic park) | Natural Park |
| Asano Ocean Breeze Park | Kokurakita | 1.61 ha | Basic Parks for Community Use (Neighborhood parks) | Urban Park |
| Katsuyama Park | Kokurakita | 9 ha | Basic Parks for City Wide Use (Comprehensive parks) | Urban Park |
| Rozanso Park | Kokurakita | 1.2 ha | Basic Parks for City Wide Use (city block parks) | Urban and Natural Park |
| Oma Bamboo Grove Park | Kokura-minami | 2.83 ha | Buffer Green Belts (Specific parks as a forest park) | Urban and Natural Park |
| Mitsutake Plum Field | Kokura-minami | 1.59 ha | Buffer Green Belts (Specific parks for agriculture) | Natural Park |
| Bijutsunomori Park | Tobata | 6.06 ha | Buffer Green Belts (Greenways) | Urban Park |
| Yomiya Park | Tobata | 8.63 ha | Basic Parks for Community Use (Community parks) | Urban and Natural Park |
| Fukuoka Kenei Central Park & Konpirayama | Tobata & Yahatahigashi | 28.24 ha | Basic Parks for City Wide Use (Comprehensive Park) | Urban and Natural Park |
| Korodai Park | Yahatahigashi | 7.67 ha | Basic Parks for City Wide Use (Comprehensive Park) | Urban Park |
| Senbonsou Park | Wakamatsu | 16.31 ha | Buffer Green Belts (Specific parks for botany) | Natural Park |
| Seita-no-mori Park | Yahatanishi | 33.82 ha | Basic Parks for City Wide Use (Comprehensive Park) | Urban and Natural Park |

* Calculated by Calcmaps.com [51] ** categorized according to MLIT, Japan [47].

*2.2. Study Case: Green Park, Kitakyushu, Japan*

2.2.1. The Climatic Character

Kitakyushu city is in Fukuoka Prefecture, Kyushu Island, Japan. Geographically, it is located at 33°53′ N and 130°53′ E of the northernmost point of Kyushu on the Kanmon Straits, separating the island from Honshu, across from the city of Shimonoseki [52]. The altitude or elevation above sea level is 6 m.

The climate of Kitakyushu city is mild, and generally warm and temperate [53]. According to the Köppen-Geiger climate classification, this climate is Cfa. The average temperature in Kitakyushu is 15.9 °C. The average temperature of August, the hottest month of the year, is 26.9 °C. January is the coldest month of the year at 5.5 °C on average. The variation in annual temperature is around 21.4 °C. Kitakyushu has four types of seasons, namely summer, autumn, winter, and spring. Summer starts at the end of June and ends in September, while autumn starts from October to November. Winter usually comes in December and disappears in February, while spring, the most beautiful season in Japan, is from March to May.

The rainfall in Kitakyushu is around 1818 mm per year, with precipitation even during the driest month. The lowest precipitation is in December, with an average of 97 mm. In June, the precipitation reaches its peak, with an average of 281 mm. The difference in precipitation between the driest and wettest months is 184 mm. The month with the highest number of rainy days is January (14.43 days). The month with the lowest number of rainy days is October (8.90 days). The month with the highest relative humidity is July (84.02%). The month with the lowest relative humidity is January (69.80%).

In August the highest number of daily hours of sunshine is measured in Kitakyushu on average. In August there is an average of 10.47 h of sunshine a day and a total of 324.58 h of sunshine throughout August. In January, the lowest number of daily hours of sunshine is measured in Kitakyushu on average. In January there are an average of 5.21 h of sunshine per day and a total of 161.64 h of sunshine. Around 2998.26 h of sunshine are counted in Kitakyushu throughout the year. On average there are 98.43 h of sunshine per month.

2.2.2. The Study Case's Location

Hibikinada Green Park is located at 1006 Takenami, Wakamatsu-ku, Kitakyushu City, Fukuoka Prefecture, Japan (see Figure 2). It has been operated from 1 April 2014, under the management of the Green Park Revitalization Consortium [54]. This place has a variety of natural landscapes and tourism attractions. It has forests, wilderness, beaches, and reservoir. There are plenty of attractions such as Pony Square, Kangaroo Square, rose garden, and tropical ecological garden (which is divided by three greenhouses). It also has outdoor stage, large lawn open space, adventure forest, Jabjabu pond, cycling terminal, ground golf, the world-longest swing, and some newest attractions, such as Bumpy open space, Dino Park, Nyoki-nyoki forest, and Fossil valley. As a supporting facility, this park also provides the Waterhouse which is a relax room with curtain-fountain, baby nursing room, and toilet. There is also an urban greening center for transmitting information on greenery and flowers, it plays a role as a consultation reception, and an exhibition and seminar holder.

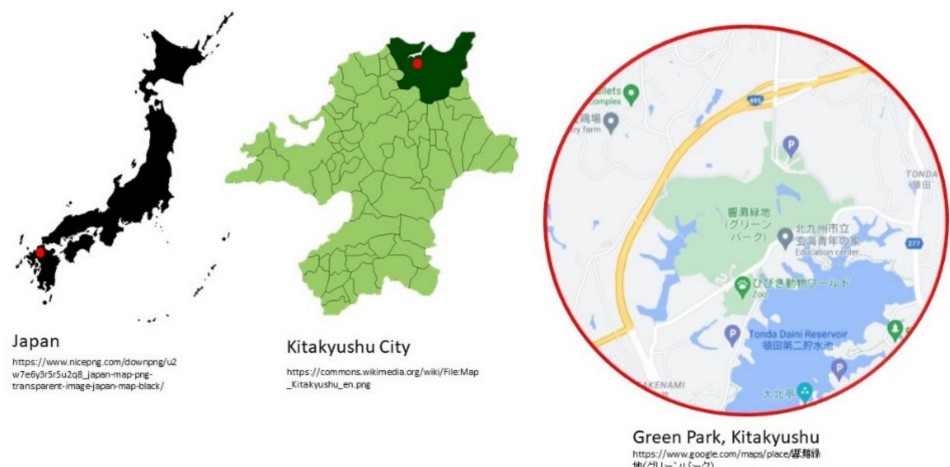

**Figure 2.** Green Park Kitakyushu: the red circle shows the location on the map.

### 2.2.3. Psycho-Ecological Condition

This study was conducted during the COVID-19 pandemic. As we know this virus outbreak began to spread since it was first reported in the city of Wuhan, China in early 2020. This may have affected the results of the study. During the COVID-19 pandemic, outdoor spaces were unsafe places for the public. However, some people prefer outdoor or open spaces, such as parks or urban forests, where they can maintain health by exercising by maintaining physical distancing compared to indoors.

### 2.3. Measurement and Methods

### 2.3.1. Population and Samples

The number of samples was obtained from the number of visitors who were willing to become respondents (answer the questionnaire) at the time the survey was conducted. The survey was conducted for approximately 2 h, between 9 and 12.30 (depending on the season and when the park gates opened). According to the Statistics Government of Canada [55], this kind of sampling method is included in volunteer sampling, where the respondents are only volunteers who must be screened (by ticket to get into this park) to get a set of characteristics suitable for the purposes of the survey. As only visitors who already have tickets are allowed to enter the Kitakyushu Green Park area, so all visitors who are already in this area and have passed the screening stage are eligible to become volunteers of this survey. Based on this sampling method, the number of samples collected is 425 people of which 187 were male, 236 were female, and 2 left the answer blank (see Table 3).

**Table 3.** Number of respondents.

| Seasons | Period | Number of Respondents | | | |
| --- | --- | --- | --- | --- | --- |
| | | Male | Female | (Blank) | Total |
| Summer | 19 July–16 August 2020 | 48 | 48 | 1 | 97 |
| Autumn | 14–18 October 2020 | 45 | 50 | 1 | 96 |
| Winter | 17 January–14 February 2021 | 47 | 70 | 0 | 117 |
| Spring | 10 April–8 May 2021 | 47 | 68 | 0 | 115 |
| | Total | 187 | 236 | 2 | 425 |

### 2.3.2. Data Collection Method

For data collection two methods were used, a micro-meteorological measurement and questionnaire survey. These two methods were conducted in Green Park, Kitakyushu, Japan. All four seasons were included; summer, autumn, winter, and spring. The surveys were conducted in four periods, summer (from 19 July to 16 August 2020), autumn (from

14 to 18 October 2020), winter (from 17 January to 14 February 2021), and spring (from 10 April to 8 May 2021).

Clothing insulation values (clo) used in these four periods were calculated using CBE Thermal Comfort Tool [56] with ASHRAE Standard 55-2020, by creating custom ensembles which are suitable with the actual season. The detailed information of clothing attributes is provided in Table 4.

**Table 4.** Clothing insulation values.

| Seasons | Gender | Clothing Insulation Value * (clo) | Attributes/Ensembles |
|---|---|---|---|
| Summer | Male | 0.50 | Typical summer indoor clothing |
| | Female | 0.45 | Bra, women's underwear, long-sleeve shirt (thin), thin skirt, and shoes or sandals. |
| Autumn | Male | 0.57 | Men's underwear, thick trousers, long sleeve shirt (thin), ankle socks, and shoes or sandals. |
| | Female | 0.56 | Bra, women's underwear, long-sleeve sweatshirt, thin skirt, ankle socks, and shoes or sandals. |
| Winter | Male | 1.55 | Men's underwear, long underwear bottoms, long underwear top, long sleeve shirt (thick), thick trousers, double-breasted coat (thick), knee socks, and shoes or sandals. |
| | Female | 1.55 | Bra, women's underwear, long underwear bottoms, long underwear top, long sleeve shirt (thick), thick trousers, double-breasted coat (thick), knee socks, and shoes or sandals. |
| Spring | Male | 0.67 | Men's underwear, T-shirt, thin trousers, single-breasted coat (thin), ankle socks, and shoes or sandals. |
| | Female | 0.84 | Bra, women's underwear, Short-sleeve dress shirt, thick skirt, single-breasted coat (thin), and shoes or sandals. |

* Calculated by CBE Thermal Comfort Tool [56].

The clothing insulation values (clo) are used to calculate PET values in the RayMan Model software. Each data unit (respondent data) uses a different clo value based on the season and gender. For example, the clo value of a male respondent in autumn is 0.57. This is based on the observation that during autumn, men tend to use clothing attributes: thick trousers, long sleeve shirts (thin), ankle socks, and shoes or sandals. Underwear (men and women) is considered as a commonly used attribute, so the underwear attribute is always counted in all conditions.

Micro-meteorological data were collected using thermal recorder and anemometer which are placed at a height of 1.2 m above ground level. In this study, an illuminance UV recorder TR-74Ui was used to record the temperature and humidity with temperature ranging from 0 to 55 °C and humidity from 10 to 95%RH [57]. For measuring wind speed, a Pro Anemometer from the HoldPeak manufacture series HP-866B-APP was used, which has a range from 0.67 to 67.1 mph (+/−5% of readings), wind temperature from −10 to 45 °C (+/−2 °C), and resolution 0.1 m/s [58]. For the Tmrt, estimated data from RayMan Model software [36,37] are used because of lack of data measurement in the investigation. The detailed information of the measurement tools is provided in Table 5.

**Table 5.** Micro-meteorological measurement tools.

| Name | Resolution | Accuracy | Output Data |
|---|---|---|---|
| UV recorder TR-74Ui | 0 to 55 °C<br>10 to 95%RH | +/−0.5 °C<br>+/−5%RH | Air Temperature (Ta)<br>Relative Humidity (RH) |
| Pro Anemometer (HoldPeak) HP-866B-APP series | 0.67 to 67.1 mph | +/−5% of readings | Wind/air velocity (v) |

The respondent data were collected by questionnaire papers. To fill out the questionnaire, collectors passively standing near the weather station asked the visitors to fill the form who were walking or by actively approaching the visitors.

2.3.3. Data Analysis Method

The data obtained both from micro-meteorological measurement and questionnaire survey are processed using a computer software. The PET value of each data unit is found by RayMan Model [36]. To calculate the PET, a unit data consisting of air temperature, (Ta), relative humidity (RH), wind velocity (v), mean radiant temperature (Tmrt), activity level/metabolic rate (M), and clothing insulation level (Icl) are needed. For the estimation of long-term studies without directly measured radiation fluxes, Tmrt can be calculated through models like RayMan [36,37]. To calculate Tmrt, the relevant properties and dimensions of the radiating surfaces and of the visible section of the sky must be known. The posture of the human body (e.g., seated or standing) is also required.

In this study, the Tmrt values are estimated by the RayMan Model software. The estimation was produced at the same time as calculating the PET value. The calculation was processed by inputting unit data from each respondent. The unit data consisted of air temperature, (Ta), relative humidity (RH), wind velocity (v), activity level/metabolic rate (M), and clothing insulation level (Icl), height, weight, age, and sex/gender. The date and time information were also included in the calculation. The geographical data of Green Park Kitakyushu location was also inputted (e.g., longitude, latitude, altitude, and time zone).

For example, a unit data of respondent no. 01 from day one of the summer season is provided in Figure 3a. This figure shows the inputted data which consisted of many variables but with no Tmrt value. The geographical data inputted are 131°12′ (longitude), 34°31′ (latitude), 6 m (altitude), and UTC + 9 (time zone). The date, time, micro-meteorological data measured, height, weight, age, sex, clothing, activity, and position data varied for each respondent. The values of the sky view factor (SVF) and horizon limitation are auto filled after the calculation is run by the software (SVF = 1 and horizon limitation = 0%, which means the complete sky is visible). Figure 3b shows the output data which produced the Tmrt estimation and PET value.

The TSV value and other questionnaire-based data are processed by Microsoft Excel and then processed into graphs for the analysis step. To analyze the relationships among variables, JMP statistical software is being used.

A regression fit model analysis with standard least squares approaches is used to understand the relationship between micro-meteorological variables and personal variables. While a correlation analysis with multivariate analysis approaches is developed to find out the correlation among personal variables. To understand the respondent tendency on some psychological questions (i.e., TSV, shading satisfaction preference, etc.), some various graphical distribution analyses are developed.

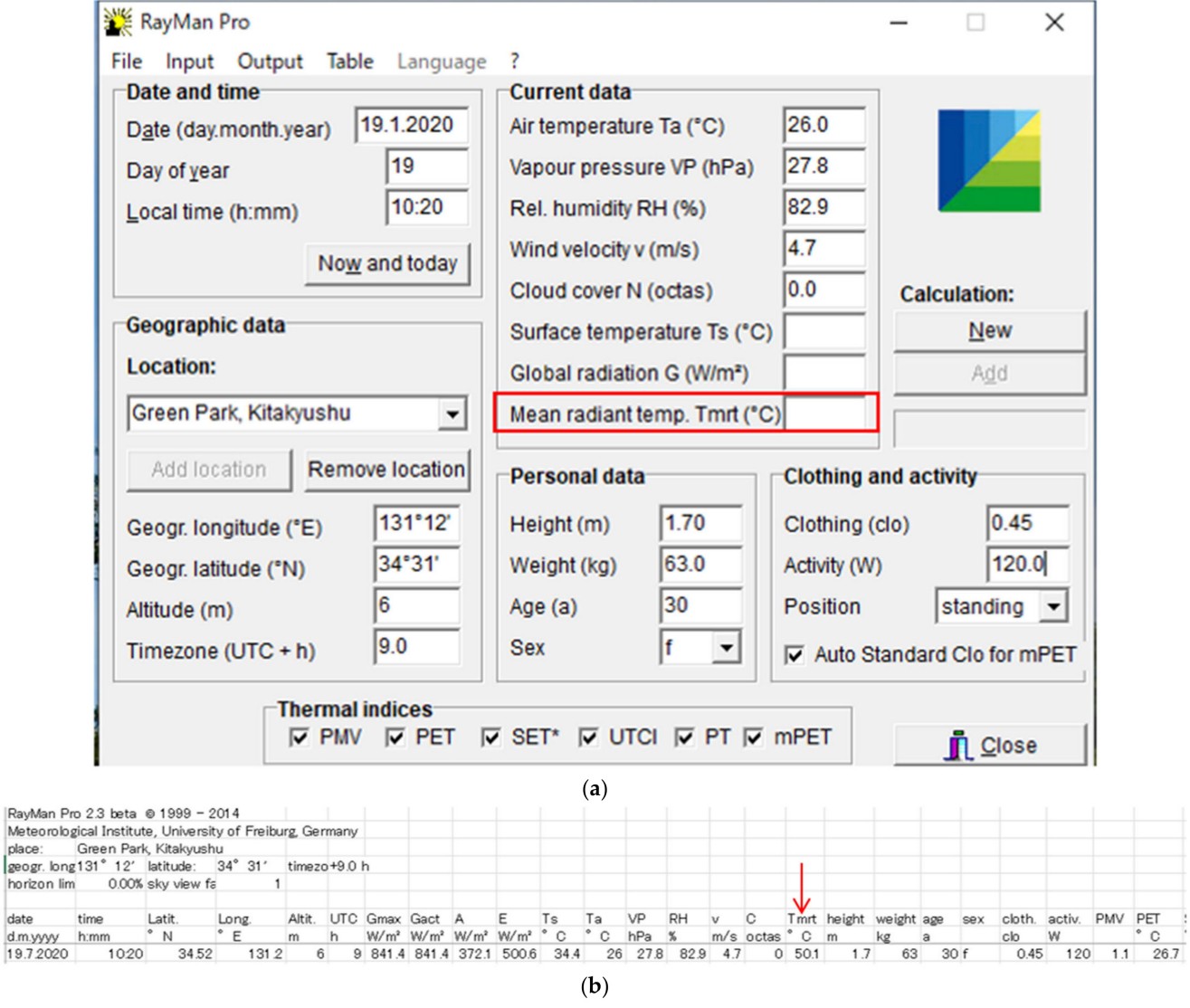

**Figure 3.** (**a**) Input data; (**b**) output data from the RayMan Model software [36,37].

### 2.3.4. Indices Used in This Study

This study used well-known indices, PET and TSV. It also introduces new indices, wind flow sensation vote (WFSV) and humidity sensation vote (HSV) to measure humidity and wind flow sensation based on respondents' vote. At the time the survey was conducted, respondents gave their opinions regarding the perceived thermal sensation, wind flow, and humidity through a questionnaire.

The form of the question is in the form of a Bipolar Likert scale, where respondents are asked to circle the answer choices on a dotted line (answer choices) that contradict each other at each end. The closer the selected point is to one end of the line (sensation), the greater the value of the sensation is felt by the respondent.

This study uses 7-point scale of TSV. The scale ranges are "cold" ($-3$), "cool" ($-2$), "slightly cool" ($-1$), "neutral" (0), "slightly warm" (+1), "warm" (+2), and "hot" (+3) conditions.

It was written on the questionnaire paper that the type of TSV and WSFV data are continuous, but HSV data type is discrete (Figures 4–6). These different types of data were a limitation of the questionnaire writing during the survey stage. So, at the time of analysis, all data types are equated to be continuous. This change in HSV data type does not affect the interpretation of respondents' answers because they have the same meaning,

counter-preference between dry (−2) and humid (+2) conditions, and the answer "just right" is a "neutral (0)" answer. The detailed information about these questions is provided in the Appendix A.

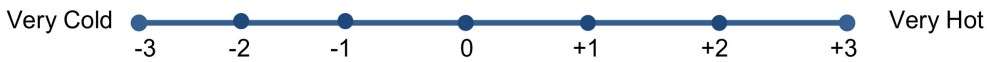

**Figure 4.** The 7-point scale of TSV.

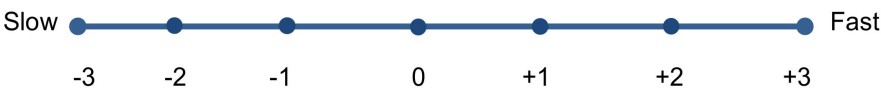

**Figure 5.** The 7-point scale of WFSV.

☐ Too dry        ☐ Slightly dry    ☐ Just right    ☐ Slightly humid    ☐ Too humid

**Figure 6.** The 5-point scale of HSV.

For the WFSV, it also uses 7-point scale, they are "very slow" (−3), "slow" (−2), "slightly slow" (−1), "normal" (0), "slightly fast" (+1), "fast" (+2), and "very fast" (+3), see Figure 5.

While, the HSV only uses 5-pont scale, they are "too dry" (−2), "slightly dry" (−1), "just right" (0), "slightly humid" (+1), and "too humid" (+2), see Figure 6.

## 3. Results

### 3.1. Respondents' Votes for Thermal, Wind Flow, and Humidity Sensation

3.1.1. Thermal Sensation Vote (TSV)

A distribution analysis has been used to understand the visitors' perception of outdoor thermal comfort in Green Park Kitakyushu (see Figure 7). Overall, most respondents feel comfortable with the thermal conditions at Green Park Kitakyushu (with a neutral sensation of 41%, slightly warm 9%, and slightly cool 9%). Especially in the spring season, the number of respondents who chose neutral was 66 (57%).

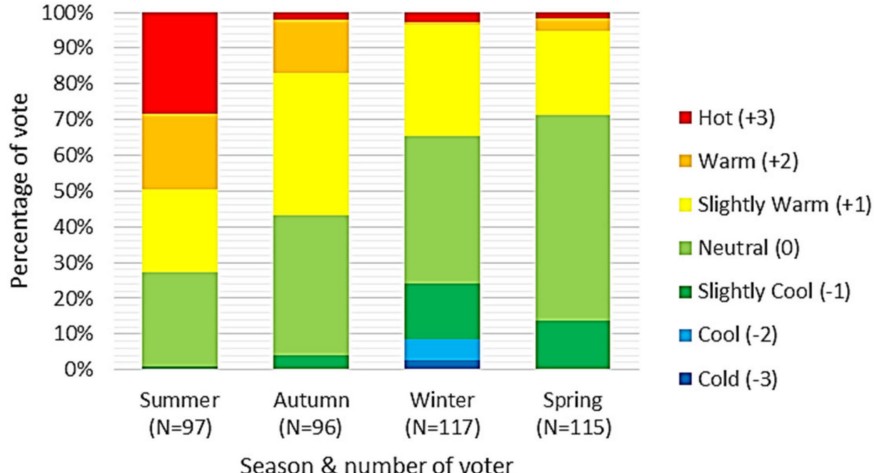

**Figure 7.** Thermal sensation vote (TSV).

During summer, although the number of respondents who chose the hot sensation (28%) was more than neutral (26%), but when viewed from the thermal comfort category where the slightly warm sensation (21%) was still comfortable. Overall, summer is still categorized as comfortable. In autumn, the most experienced thermal sensation by respondents was slightly warm (40%), followed by neutral (39%). While in winter, most feel

neutral (41%). It is interesting to observe that in winter, only a small number of respondents chose cool and cold answers. This may invite the next question, what variables have the most influence on the answer, whether the variables are personal variables (e.g., clothing insulation or activity level) or environmental conditions (temperature, humidity, wind, and radiant temperature). The quality of shading (both from buildings and vegetation) may also affect the response to thermal sensation (TSV).

### 3.1.2. Wind Flow Sensation Vote (WFSV)

In the WFSV question, respondents are asked to determine their tendency of sensation to air movement that is felt around their place (see Figure 8). Overall, most respondents feel that the wind around them is neutral (44%) or the wind speed is moderate (not fast and not slow). Meanwhile, when comparing the four seasons, according to respondents, the season with the most neutral wind speed is autumn (50%), followed by winter (45%), and spring (44%).

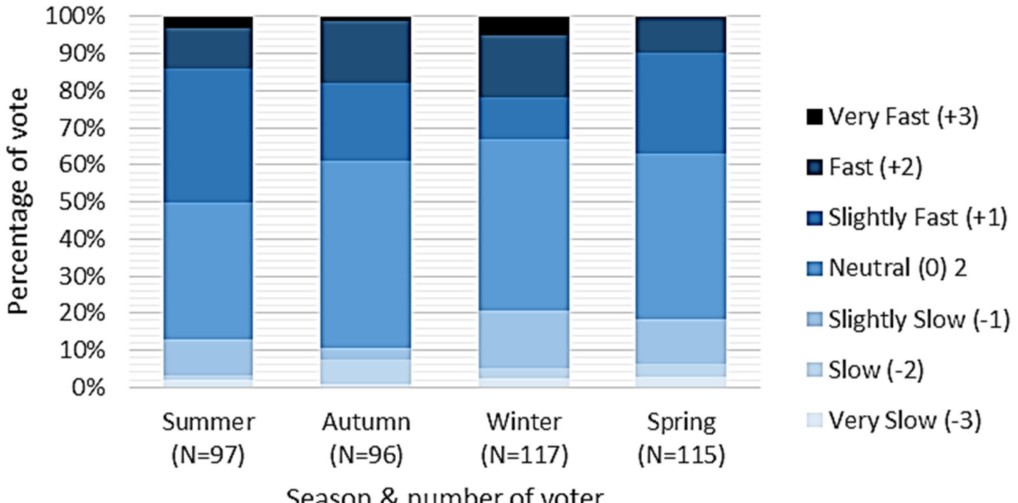

**Figure 8.** Wind flow sensation vote (WFSV).

The season that feels the most uncomfortable with high wind speeds is winter, with the percentage of Fast and Very fast voters being 16% and 5%, respectively. Then followed by autumn, namely Fast (17%) and Very fast (1%).

### 3.1.3. Humidity Sensation Vote (HSV)

Regarding the air humidity felt by visitors when the survey was carried out (see Figure 9), broadly most of the respondents (56%) answered Just Right (do not feel the sensation of moist or dry). Of the five answer choices, there are two categories based on comfort, namely the comfortable category (consisting of Slightly Dry, Just Right, and Slightly Humid) and the uncomfortable category (Too Dry and Too Humid). Based on this category, most visitors (83%) feel comfortable with the humidity conditions in the Green Park Kitakyushu.

Viewed from the season period, the highest number of respondents who feel Slightly Humid and Humid sensation is summer, with a percentage of 39% and 16%, respectively. While the highest percentage for the sensation of neutral humidity (Just Right) is winter, which is 68%.

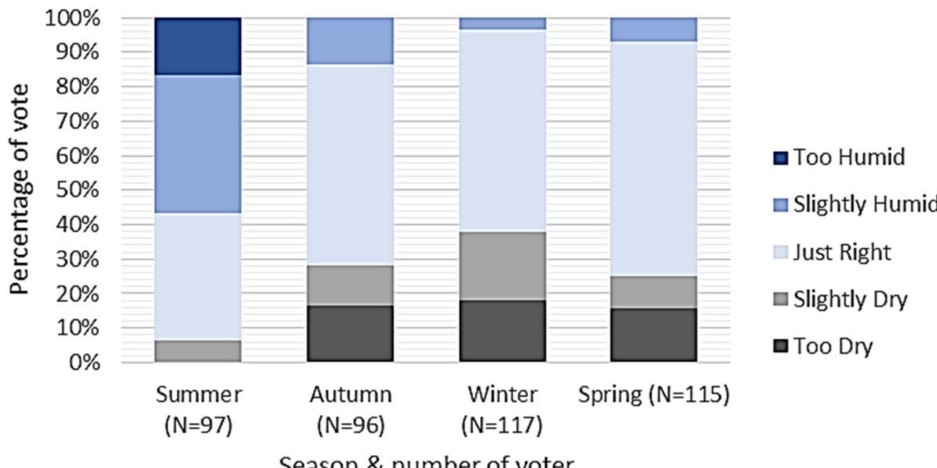

**Figure 9.** Humidity sensation vote (HSV).

*3.2. Acceptability and Satisfaction Level of Thermal Comfort*

3.2.1. Thermal Acceptability

As shown in Figure 10, most respondents (84%) can accept the thermal conditions in the Green Park environment. If observed further, only summer has a slight difference between the number of respondents who can accept (56%) and who cannot accept (42%) the thermal conditions of their environment. Meanwhile, the other three seasons (autumn, winter, and spring) have significant differences in the number of voters (thermally acceptable >90% and not acceptable <10%).

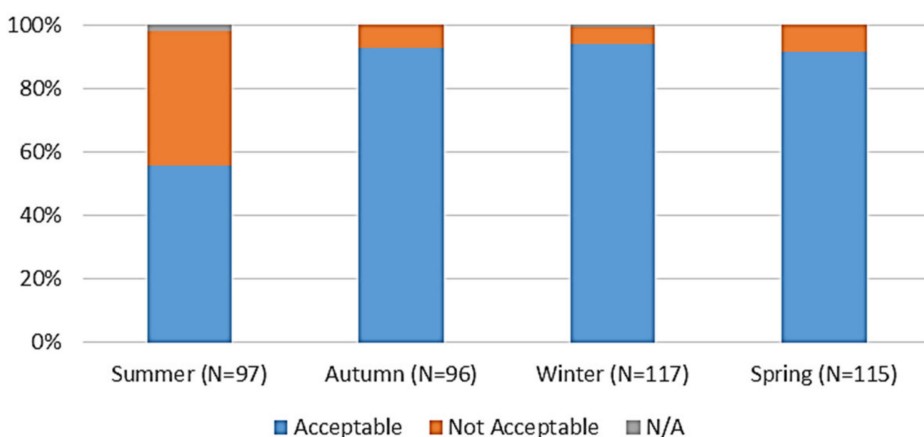

**Figure 10.** Thermal acceptability.

3.2.2. Thermal Satisfaction Level

Overall, there were two most answers regarding the level of satisfaction with the thermal environment at the time this survey was conducted, namely Just like this (49%) and Cooler is better (41%) (see Figure 11). The interesting thing about the results of this survey is that in winter, the number of voters who answered Cooler is better (28%) and was higher than that of Warmer is better (19%). This begs the question whether there are other factors that cause respondents to have such a level of satisfaction. Although when compared to other seasons, the highest number of voters for Warmer is better is in winter (summer 2%, autumn 4%, and spring 9%).

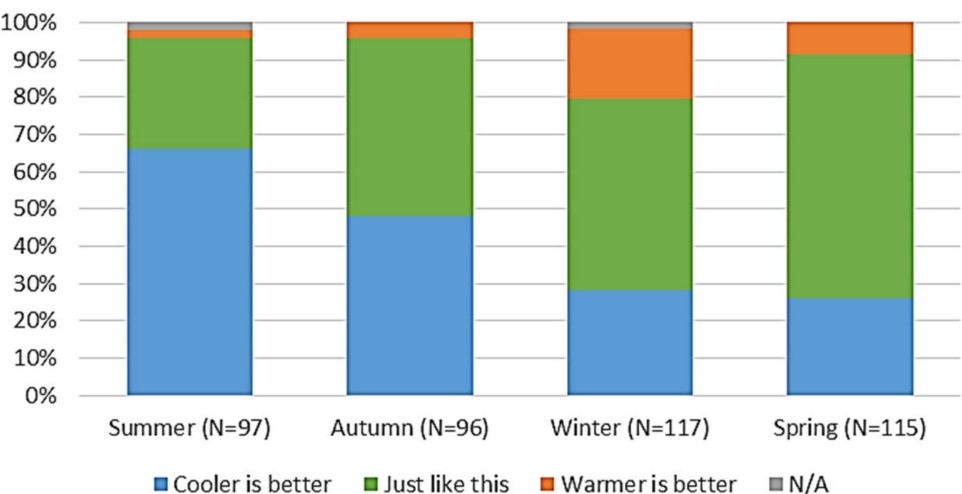

**Figure 11.** Thermal satisfaction level.

The highest voter for Cooler is better was in summer (66%), then followed by autumn (48%). This certainly shows that in summer and autumn, the thermal conditions of the Green Park environment are relatively hotter than in winter and spring.

### 3.3. Satisfaction Preferences for Shading, Sunlight, and Wind Conditions

3.3.1. Shading Satisfaction Preferences

In summer, respondents were dissatisfied with the existing shading conditions, most of them felt Need more shading (75%). In the autumn season, most of the respondents also answered Need more shading (54%), while those who answered Fit right were 46%, and no one answered Need less shading (0%). In winter, most chose Fit right (66%), followed by Need more shading (32) and Need less shading (2%). For the spring season, the results are relatively the same as in summer and autumn, where most of them answered Need more shading. Thus, it is only in winter that voters are most satisfied with the shading conditions in the Kitakyushu Green Park environment.

Overall (see Figure 12) most of the respondents were dissatisfied and needed more shading than was available at the time the survey was conducted. However, if observed in Figure 6, the difference in the percentage of respondents who are not satisfied (Need more shading) and satisfied (Fit right) is not so significant, which is only 7%.

3.3.2. Sunlight Satisfaction Preferences

The results of the survey on the question of respondents' satisfaction preferences for the presence of sunlight in the Green Park environment (see Figure 13) showed that visitors were satisfied (Fit right), with an overall percentage of 83%. Among the four seasons, in summer the most respondents chose Need less sunlight (16%). While other seasons are the opposite, more people choose to need more sunlight than need less sunlight.

3.3.3. Wind Satisfaction Preferences

Like the results of the previous survey on sunlight, the satisfaction preference for wind conditions in the Green Park environment is dominated by Fit right answers (with an overall percentage of 72%). The number of respondents who chose need more wind over need less wind was summer and autumn, with a percentage ratio of 28% versus 9% and 9% versus 8%, respectively). Whereas in the opposite situation, winter and spring have a higher percentage of voters who need less wind than need more shading, with a percentage ratio of 16% versus 7% and 22% vs. 11%, respectively. Overall, the respondents were satisfied with the wind conditions in the Green Park environment, especially in the location where this survey was conducted (see Figure 14).

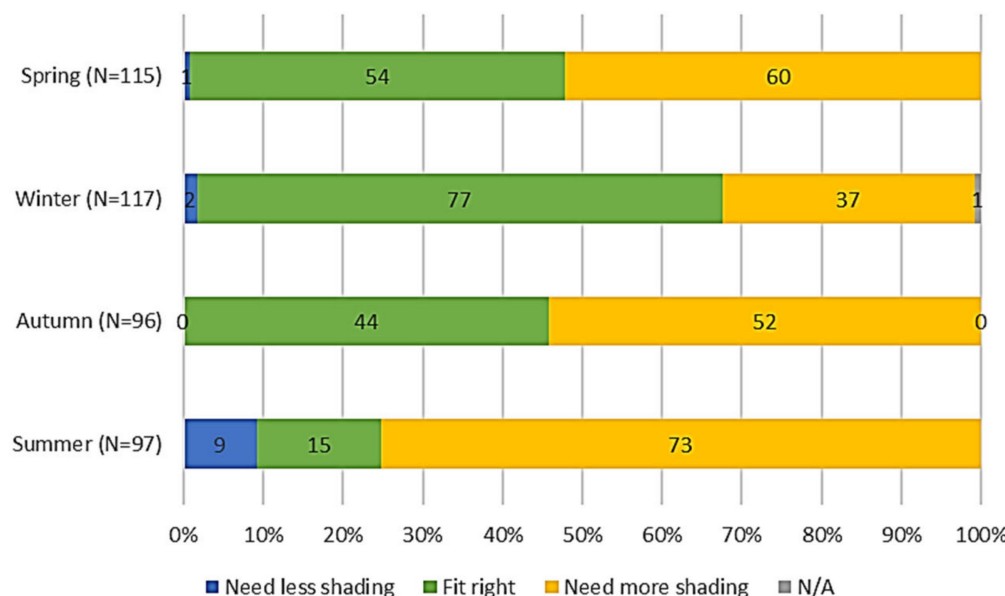

**Figure 12.** Shading satisfaction preferences.

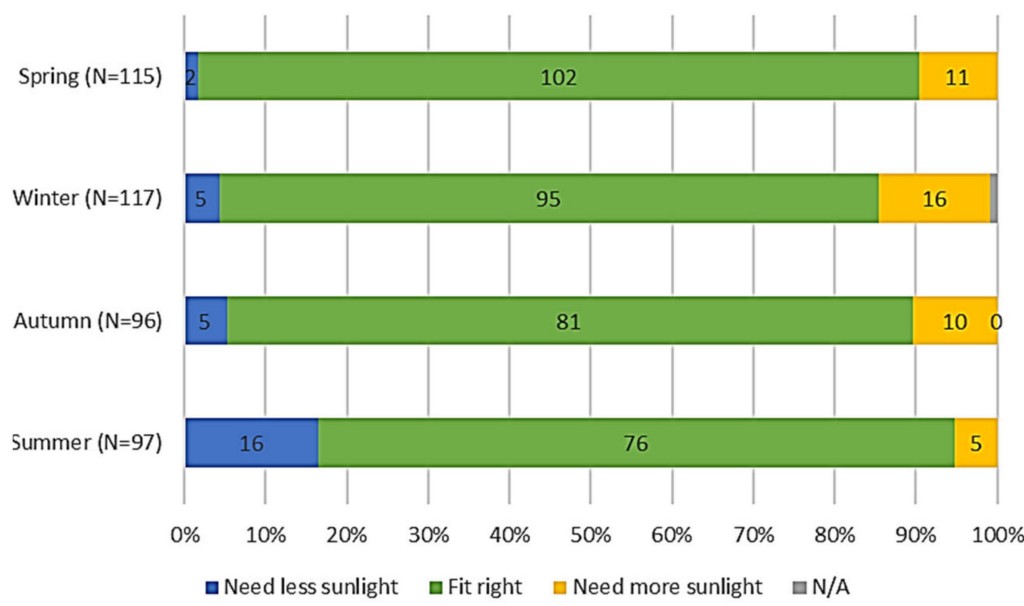

**Figure 13.** Sunlight satisfaction preferences.

*3.4. The Relationships between Micro-Meteorological and Personal Variables*

3.4.1. The Most Significant Micro-Meteorological Variable of PET

To understand the relationship between PET and the micro-meteorological variables, a regression analysis is used by applying the Fit Model method with the Standard Least Squares approach (see Figure 15). There were five variables analyzed, namely PET, air temperature (Ta), relative humidity (RH), air velocity (v), and mean radiant temperature (Tmrt). All the variables' data have been standardized before analysis by JMP statistical software. This is done to maintain the equality of the values of the five variables analyzed.

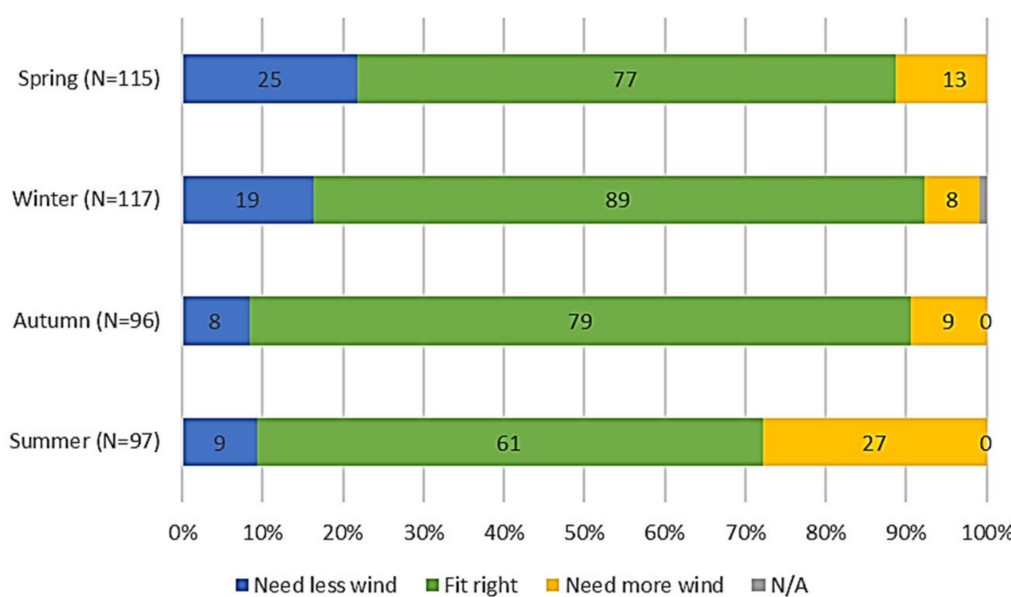

**Figure 14.** Wind satisfaction preferences.

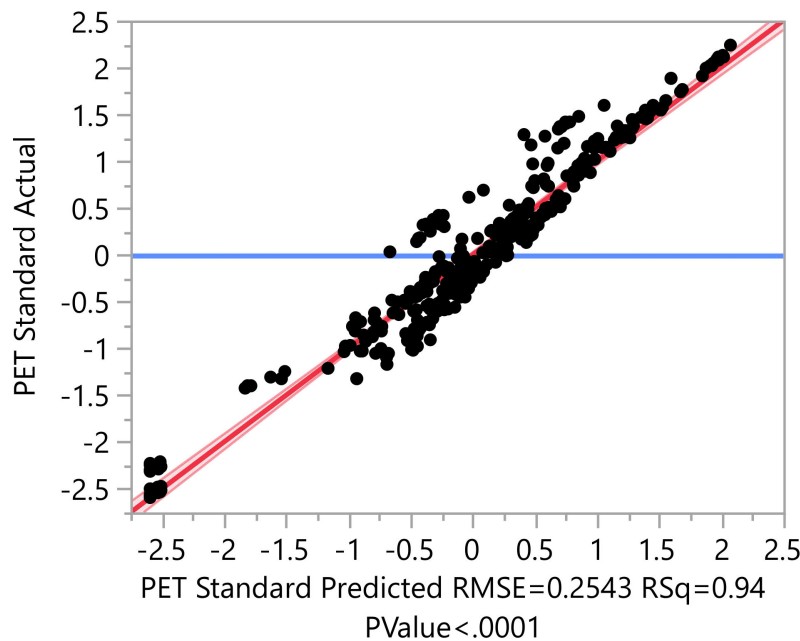

**Figure 15.** A relationship between PET and micro-meteorological variables.

Based on the results, the value of reliability ($R^2$) of the correlation of the five variables is 0.94 (close to 1), so the data used are reliable or accurate to be used as material for analysis in a study. The significance value (*p* value) is <0.0001 (close to 0), meaning that it is significant, or in other words, the chances of this finding being missed are almost non-existent.

Based on the parameter estimates above (Table 6), an equation can be drawn up as follows:

$$PET = 0.26\,Ta - 0.1\,RH - 0.16\,v + 0.65\,Tmrt \tag{1}$$

**Table 6.** Parameter estimates between PET and micro-meteorological variables.

| Term | Estimate | Std Error | t Ratio | Prob >|t| |
|---|---|---|---|---|
| Intercept | 0.0008044 | 0.012397 | 0.06 | 0.9483 |
| Ta Standard | 0.263544 | 0.053904 | 4.89 | <0.0001 * |
| RH Standard | −0.10604 | 0.015478 | −6.85 | <0.0001 * |
| v Standard | −0.168424 | 0.017673 | −9.53 | <0.0001 * |
| Tmrt Standard | 0.6577773 | 0.060381 | 10.89 | <0.0001 * |

* $p$ value is significant.

The most influencing environmental factor to the PET value is mean radiant temperature (Tmrt) (see Equation (1)). Its positive relationship (0.65) means the higher the Tmrt value, the higher the PET value.

Based on Figure 16, it can be seen the type of relationship between PET and micrometeorological variables. Factors that are positively related are the temperature variable (Ta), and the mean radiant temperature (Tmrt) variable, which means the higher the value of Ta and Tmrt, the higher the PET value. On the other hand, the relation value of air velocity (v) and RH variables are negative, meaning that the smaller the value, the higher the PET value.

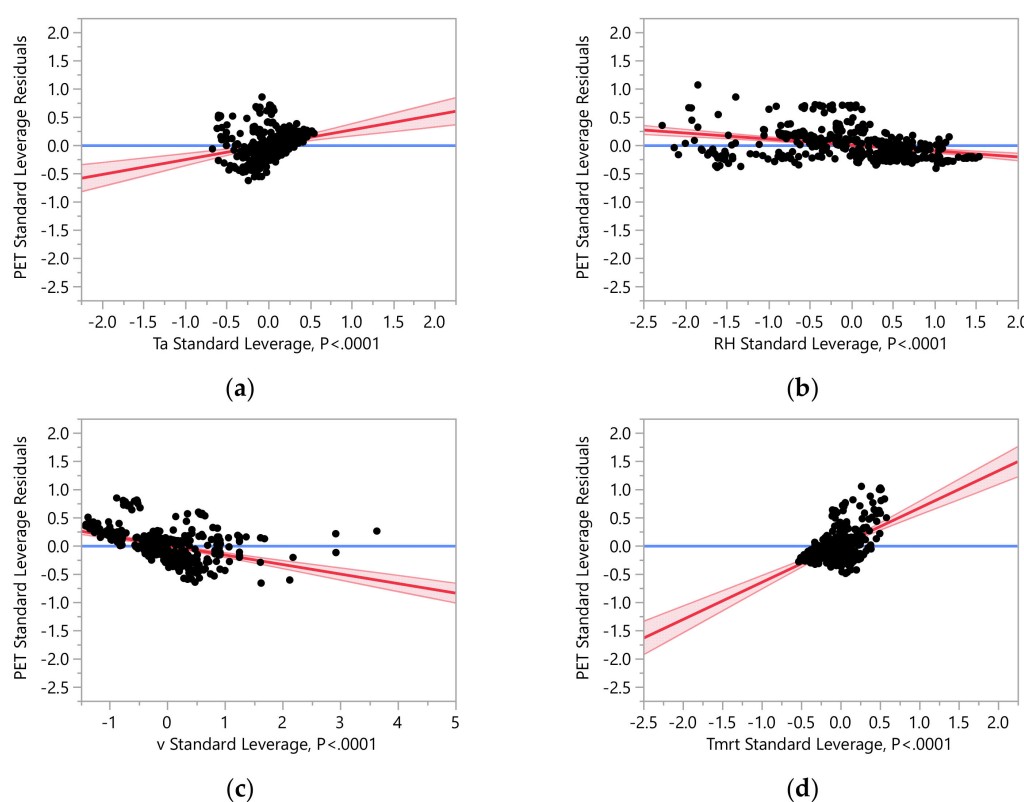

**Figure 16.** Relationships between PET standard and micro-meteorological variables: (**a**) PET and Ta; (**b**) PET and RH; (**c**) PET and v; (**d**) PET and Tmrt.

### 3.4.2. The Relationship between Micro-Meteorological Variables and TSV

The relationship between micro-meteorological variables (Ta, RH, v, and Tmrt) and TSV is analyzed by regression analysis method, with Fit Model approach and Standard Least Squares personality. Based on the results of the analysis (Figure 17), the reliability value ($R^2$) of the relationship between TSV and the four micro-meteorological variables is 0.30 (far from 1 and less than the recommended minimum 0.6)), so the data used is not reliable or not accurate to be used as material for analysis in a study. However, the

significance value (*p* value) is <0.0001 (close to 0), meaning that it is significant, or in other words, the chances of this finding being missed are almost non-existent.

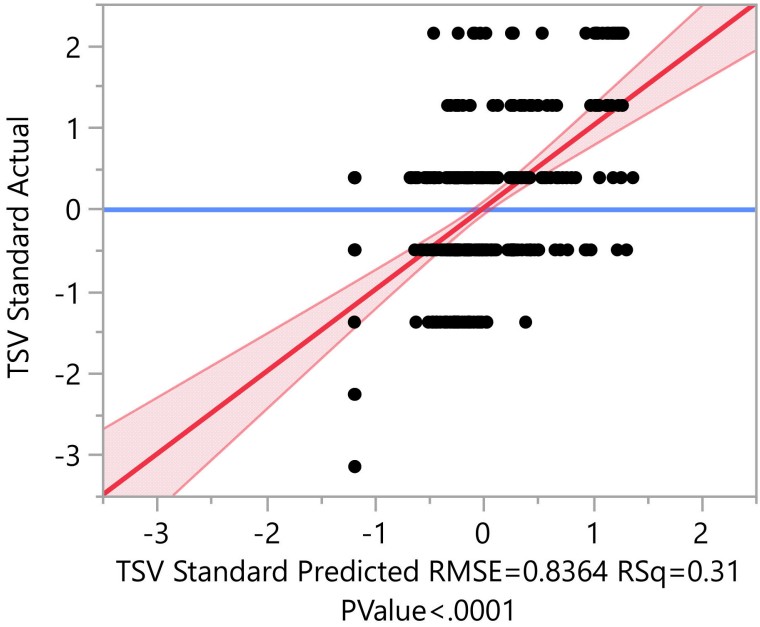

**Figure 17.** A relationship between TSV standard actual and predicted.

The relationship between micro-meteorological variables (Ta, RH, v, and Tmrt) and TSV, is shown in Table 6.

Based on the parameter estimates table (Table 7), an equation can be drawn up as follows:

$$TSV = 0.64\ Ta - 0.05\ RH + 0.02\ v - 0.07\ Tmrt \tag{2}$$

**Table 7.** Parameter estimates between TSV and micro-meteorological variables.

| Term | Estimate | Std Error | t Ratio | Prob>|t| |
|---|---|---|---|---|
| Intercept | 0.0018202 | 0.040768 | 0.04 | 0.9644 |
| Ta Standard | 0.6432868 | 0.17727 | 3.63 | 0.0003 * |
| RH Standard | −0.050148 | 0.050901 | −0.99 | 0.3251 |
| v Standard | 0.0218456 | 0.058119 | 0.38 | 0.7072 |
| Tmrt Standard | −0.079883 | 0.198571 | −0.40 | 0.6877 |

* *p* value is significant.

It can be seen from Equation (2), the environmental factor that most influences the TSV value is Ta (air temperature), with a positive relationship (0.64). In other words, the greater the Ta value, the greater the TSV value. This shows that according to the respondents' perception, the most influential factor on the value of thermal comfort in the Kitakyu-shu Green Park environment is temperature conditions. The relationship between the TSV variable and the Ta and v variables is positive, while the RH and Tmrt variables are negative.

3.4.3. The Relationship between Micro-Meteorological Variables and WFSV

Based on the results of the analysis (Figure 18), the reliability value ($R^2$) of the relationship between WFSV and the four micro-meteorological variables is 0.02 (very far from 1 and less than the recommended minimum 0.6), so the data used is very unreliable or inaccurate for analysis. Likewise, the significance value is 0.12 (>0.1), meaning that it is not significant, or in other words there is a 12% chance that these findings are wrong.

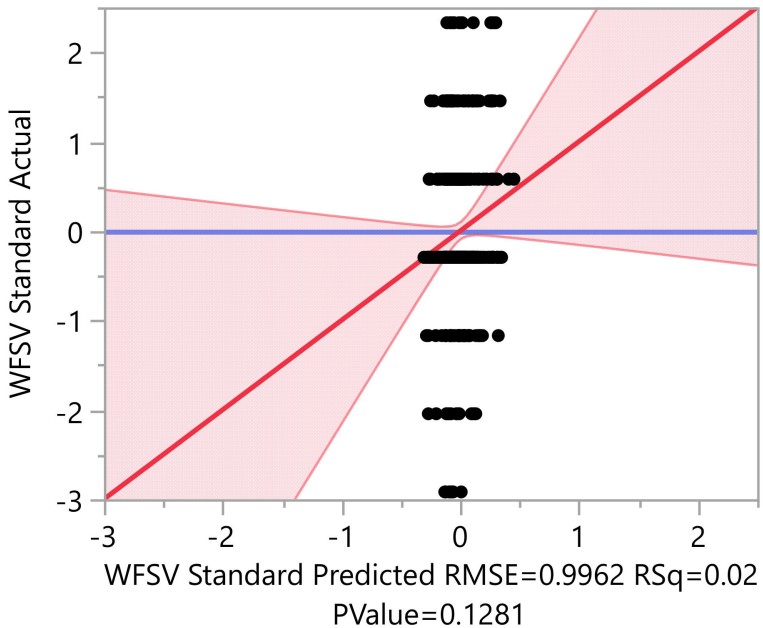

**Figure 18.** A relationship between WFSV standard actual and predicted.

However, the correlation between micro-meteorological variables (Ta, RH, v, and Tmrt) and WFSV is shown in Table 7.

Based on the parameter estimates table above (Table 8), an equation can be drawn up as follows:

$$\text{WFSV} = 0.39\,\text{Ta} - 0.02\,\text{RH} + 0.005\,\text{v} - 0.32\,\text{Tmrt} \tag{3}$$

**Table 8.** Parameter Estimates between WFSV and micro-meteorological variables.

| Term | Estimate | Std Error | t Ratio | Prob > \|t\| |
|---|---|---|---|---|
| Intercept | 0.001687 | 0.048732 | 0.03 | 0.9724 |
| Ta Standard | 0.3932379 | 0.213465 | 1.84 | 0.0662 |
| RH Standard | −0.023098 | 0.061511 | −0.38 | 0.7075 |
| v Standard | 0.0054662 | 0.071915 | 0.08 | 0.9394 |
| Tmrt Standard | −0.323141 | 0.239887 | −1.35 | 0.1787 |

The most influential environmental factor on the WFSV value (see Equation (3)) is air temperature (Ta), with a positive relationship (0.39). In other words, the greater the Ta value, the greater the WFSV value. This shows that according to the respondent's perception, the most influential factor on the sensation of wind flow in the Kitakyushu Green Park environment is temperature. The relationship between variables that has a positive value is between WFSV with Ta and v, while the negative value is between WFSV with variables RH and Tmrt.

### 3.4.4. The Relationship between Micro-Meteorological Variables and HSV

The regression analysis result between HSV and the four micro-meteorological variables (Figure 19) shows that the value of reliability ($R^2$) is 0.22, so the data used are very unreliable or very inaccurate for analysis. However, the significance value is <0.0001 (close to 0), meaning that it is significant, or in other words, the chances of this finding being missed are almost non-existent.

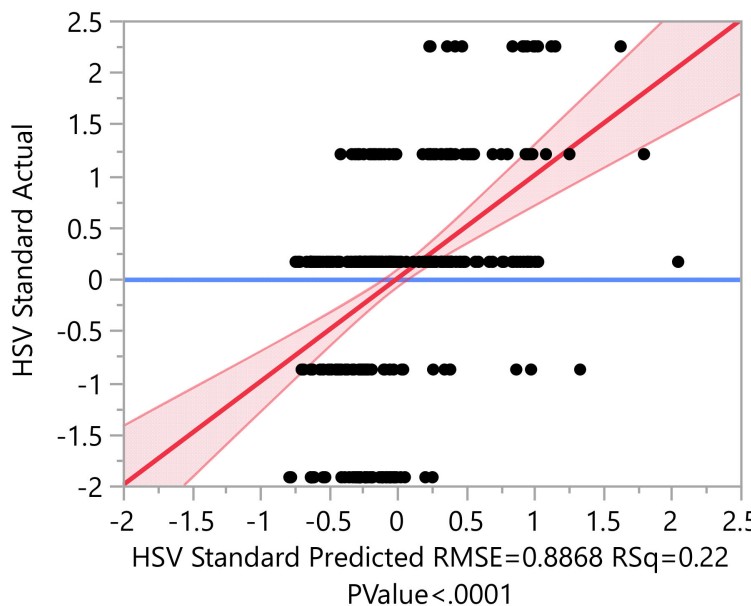

**Figure 19.** A relationship between HSV standard actual and predicted.

However, the results of the estimated correlation between micro-meteorological variables (Ta, RH, v, and Tmrt) and HSV are shown in Table 8.

According to the parameter estimation results between HSV and micro-meteorological variables (Table 9), an equation can be drawn up as follows:

$$HSV = 0.36\ Ta - 0.06\ RH + 0.24\ v + 0.15\ Tmrt \tag{4}$$

**Table 9.** Parameter estimates between HSV and micro-meteorological variables.

| Term | Estimate | Std Error | t Ratio | Prob > \|t\| |
|---|---|---|---|---|
| Intercept | 0.0011969 | 0.043221 | 0.03 | 0.9779 |
| Ta Standard | 0.3628165 | 0.187936 | 1.93 | 0.0542 |
| RH Standard | −0.066378 | 0.053964 | −1.23 | 0.2194 |
| v Standard | 0.2413083 | 0.061616 | 3.92 | 0.0001 * |
| Tmrt Standard | 0.1568445 | 0.210519 | 0.75 | 0.4567 |

* *p* value is significant.

Based on the HSV Equation (4), the HSV value is also strongly influenced by the value of Ta (air temperature), with a positive correlation (0.36). In other words, the HSV value will increase as the Ta value increases. This shows that according to the respondent's assessment, the most influential variable on the sensation of humidity in the Kitakyushu Green Park environment is air temperature conditions. In addition to air temperature, a positive relationship is between Tmrt and v, while a negative relationship is RH.

3.4.5. The Relationship between PET and Personal Variables (TSV, WFSV, and HSV)

The multivariate analysis result shows the correlations between the four variables (PET, TSV, WFSV, and HSV). The correlations are estimated by pairwise method (see Table 10).

**Table 10.** Pairwise correlations between PET and personal variables.

| Variable | by Variable | Correlation | Signif Prob | | | |
|---|---|---|---|---|---|---|
| TSV Standard | PET Standard | 0.5095 | <0.0001 * | | | |
| HSV Standard | PET Standard | 0.3407 | <0.0001 * | | | |
| HSV Standard | TSV Standard | 0.2580 | <0.0001 * | | | |
| WFSV Standard | TSV Standard | 0.1020 | 0.0372 * | | | |
| WFSV Standard | PET Standard | 0.0409 | 0.4041 | | | |
| HSV Standard | WFSV Standard | −0.0690 | 0.1589 | | | |

\* *p* value is significant.

Based on the table, the strongest relationship in the four variables is between TSV and PET. The correlation coefficient between TSV and PET is 0.5 (positive correlation). It means that both variables move in the same direction or when the PET value is high, the TSV value is also high. The correlation coefficient between PET and HSV also indicates a positive relationship (0.34). In the contrary, this table indicates that there is no relationship between PET and WFSV.

## 4. Discussion

As mentioned in the previous chapter on the results of the TSV, it is interesting to observe that, in winter, only a small proportion of respondents chose Cool and Cold answers. This means that most people do not feel cool when they are outside, especially in Green Park Kitakyushu. If we return to the results of the regression analysis between TSV and micro-meteorological variables, it was found that the most influencing factor for thermal comfort is air temperature. Therefore, logically in winter people will choose a cool or cold sensation. But in this result, it is the opposite. According to Velt and Daanen [43], people feel more uncomfortable because their mean body temperature is lower than ideal. Then most likely there are other factors that cause it.

The first possibility is because in winter people wear the right clothes for outdoor activities. According to De Carli [59], people tend to dress appropriately when they know they will be in cold outdoor conditions, to a large extent, the temperature outside at 6 am affects people's clothing choices.

The second possible reason is the role of activity level (metabolism rate) in a person's decision to choose which environmental thermal conditions are more suitable with the thermal conditions felt by the body. Typically, core body temperature is elevated when we face continuous whole-body work and exercise [60]. The many choices of attractions and play facilities offered at Green Park could increase one's activity level. Then, this high activity level affects choosing a suitable thermal sensation for the body temperature.

The most influencing environmental factor to the PET value is mean radiant temperature (Tmrt) with a positive relationship. It means the higher the Tmrt value, the higher the PET value. According to Tan [15], the Tmrt value is influenced by how much shading is produced from the presence of vegetation or buildings around the measurement location. This shows that the presence of shadows greatly affects the thermal comfort value in the Green Park Kitakyushu. This finding strengthens the previous studies that show the important role of shading in cooling urban temperatures.

Based on the regression analysis results between micro-meteorological variables and personal variables, there lacks reliability values. These might be because of the adequacy of the number of data units, the timeliness between recording micro-meteorological measurement data and the questionnaire, or accuracy in preparing research methods and plans. Future research can be developed by increasing the number of visitor participation (respondents), so that research results can be more accurate and develop a more detailed and measurable research plan.

## 5. Conclusions

The first objective of this study is to understand the people's perception of outdoor thermal sensation (TSV), wind flow sensation (WFSV), and humidity sensation (HSV). It found that most of respondent were feeling comfort with the thermal, wind, and humidity condition. The sensation of thermal and the wind flow were mostly neutral, and the sensation of humidity were also in the mid-range (just right, nor humid and dry).

The acceptability and satisfaction level of thermal comfort were positive. Most of respondents accepted and were satisfied with the thermal condition. For the satisfaction preferences for shading, most of the respondents in three seasons (summer, autumn, and spring) were dissatisfied with the actual shading condition and agreed to gain more shading, to get more chance for shelter from the hot sun. Only respondents of winter season were mostly feeling satisfied. For the sunlight and wind satisfaction preferences, most of respondents in all seasons were feeling satisfied with the actual condition, no compliment.

The main objective is to understand the relationships between micro-meteorological and personal variables of outdoor thermal comfort. The most significant micro-meteorological variable for the PET value is mean radiant temperature (Tmrt). As the Tmrt value is influenced by how much shading is produced from the presence of vegetation or buildings around the measurement location, this finding shows that the shadow was very important to the thermal comfort conditions of the Green Park Kitakyushu.

The most influential micro-meteorological variable for the three different personal variables (TSV, WFSV, and HSV) is air temperature. The strongest relationship between the four variables is between TSV and PET. The correlation coefficient between TSV and PET is 0.5 (positive correlation).

## 6. Limitations

The mean radiant temperature (Tmrt) data in this study are estimated by a computer software and has not been measured in the field investigation. The number of data units is relatively small according to the result of the regression analysis which is shown by the small value of reliability ($R^2$). The lack of these information may affect the results of the study.

**Author Contributions:** Conceptualization, D.H. and B.J.D.; methodology, D.H. and M.D.K.; software, D.H.; validation, B.J.D. and M.D.K.; formal analysis, D.H.; investigation, D.H.; resources, D.H.; data curation, D.H.; writing—original draft preparation, D.H.; writing—review and editing, D.H.; visualization, D.H.; supervision, B.J.D. and M.D.K.; project administration, D.H.; funding acquisition, D.H. All authors have read and agreed to the published version of the manuscript.

**Funding:** This research was funded by the Ministry of Education, Culture, Sports, Science and Technology of Japan (MEXT) Scholarship.

**Institutional Review Board Statement:** Not applicable.

**Informed Consent Statement:** Not applicable.

**Data Availability Statement:** The data presented in this study are available on request from the corresponding author. The data are not publicly available due to privacy matter.

**Acknowledgments:** This work was supported by the Ministry of Education, Culture, Sports, Science and Technology of Japan (MEXT) Scholarship, and The University of Kitakyushu, Japan. We also thank to the management of Green Park Kitakyushu for allowing us doing the field surveys and providing data for this study. Great appreciation also addressed to respondents who voluntary answering our questionnaire during the surveys. Thanks also to our colleague who help us to spread the questionnaire to the respondents.

**Conflicts of Interest:** The authors declare no conflict of interest. The funders had no role in the design of the study; in the collection, analyses, or interpretation of data; in the writing of the manuscript, or in the decision to publish the results.

**Appendix A**

The questionnaire is divided into three parts: A, B, and C. The A part is questions about personal information concluding gender, age, height, weight, nationality, and the current city he/she lives. The B part is questions about frequency and reason to visit the urban park, favorite season and area, and their expectations for the facilities in the future. The C part is preferences and satisfactions about outdoor thermal comfort. The questions about three indices (TSV, WFSV, and HSV) are written on the C part (Figure A1).

**A. Basic Personal Information:**

1) Gender :☐ Male   ☐Female
2) Age    :☐10's   ☐20's  ☐30's  ☐40's  ☐50's  ☐60's    ☐70's plus
3) Height :              cm      weight :         kg
4) Nationality :
5) The current city (you live) :
   ☐ Inside Kitakyushu*                              ☐ Other prefecture (In Japan)
   ☐ In Fukuoka prefecture                          ☐ Abroad (Outside Japan)
   *) If you live in Kitakyushu city, how long have you been living here?
   ☐Less than 1 year                                 ☐ Between 3 and 10 years
   ☐Between 1 and 3 years                          ☐ More than 10 years

**B. Questions about This Park**

1) **Please tell us what your reason to visit this park today? Please circle (O) one or more from the list below.**

| 1A | Fitness or doing sports |
| 1B | Play with children |
| 1C | Have a picnic or gather with my friends |
| 1D | For educational purpose |
| 1E | For a pleasant diversion |
| 1F | For a community event |

2) **Please tell us what your reason to visit this park on the previous days? Please circle (O) one or more from the list below. If this is your first time, please leave it blank.**

| 1A | Fitness or doing sports |
| 1B | Play with children |
| 1C | Have a picnic or gather with my friends |
| 1D | For educational purpose |
| 1E | For a pleasant diversion |
| 1F | For a community event |

3) **How often do you come here?**

☐ Daily or more often                    ☐ Once or twice a year
☐ Weekly or more often                  ☐ This is my first time
☐ Monthly or more often

4) **Please choose the season you like most in this park.**
☐ Summer                                  ☐ Winter
☐ Autumn                                  ☐ Spring

5) **How important is this park for you? Please circle (O) one of this nodes.**

Very Not Important ●———●———●———●———●———● Very Important
                    -3   -2   -1    0   +1   +2   +3

**Figure A1.** *Cont.*

6) **Which your favourite area(s) of this Park? Please check (v) one or more.**
   - ☐ The lawn square
   - ☐ Inside the building (indoor area)
   - ☐ Near the building (outdoor area)
   - ☐ Near the flowers and trees (natural area)
   - ☐ Playground (kid's area)

7) **What do you want to be available in the future on this park? (Your expectations)**
   - ☐ Water play facilities (swimming pool, water fountain, etc.)
   - ☐ Pets play facilities (for dogs, cats, etc.)
   - ☐ More animals varieties (like a zoo)
   - ☐ Camping space (Picnic, Barbeque, etc.)
   - ☐ Can stay all the night (to see the stars, etc.)
   - ☐ Athletic ground or sports space (baseball, dodgeball, softball, soccer, badminton, etc.)
   - ☐ Skate Park (for Skateboard)
   - ☐ Others .........................................
   - ☐ I am satisfied with current condition.

C. **Questions about Thermal Comfort**

1) **How do you feel about the temperature at this moment? Please circle (O) one of this nodes.**

| | Temperature | |
|---|---|---|
| Very Cold | ●——●——●——●——●——●——● | Very Hot |
| | -3  -2  -1  0  +1  +2  +3 | |

2) **How do you feel about the air flow at this moment? Please circle (O) one of this nodes.**

| | Air Flow | |
|---|---|---|
| Slow | ●——●——●——●——●——●——● | Fast |
| | -3  -2  -1  0  +1  +2  +3 | |

   **2a) Do you feel being disturbed by the air flow?**

   ☐ Yes        ☐ No

3) **How do you feel about humidity at this moment?**

   ☐ Too dry     ☐ Slightly dry   ☐ Just right     ☐ Slightly humid     ☐ Too humid

4) **Are you satisfied with the current outdoor environment?**

   ☐ Yes        ☐ No

6) **Which the outside temperature that suitable for you?**

   ☐ Cooler is better   ☐ Just like this        ☐ Warmer is better

7) **How satisfied are you with the temperature in here? Please circle (O) one of this nodes.**

| | | |
|---|---|---|
| Very Disappointed | ●——●——●——●——●——●——● | Very Satisfied |
| | -3  -2  -1  0  +1  +2  +3 | |

**Figure A1.** *Cont.*

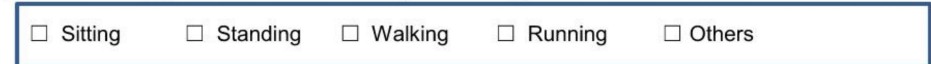

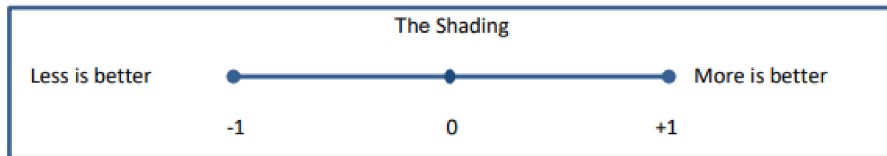

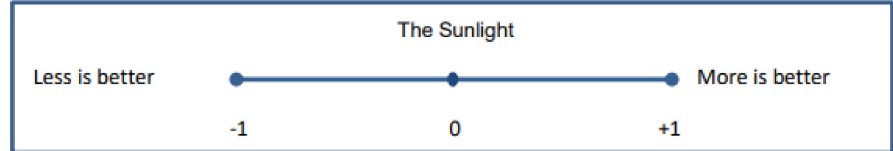

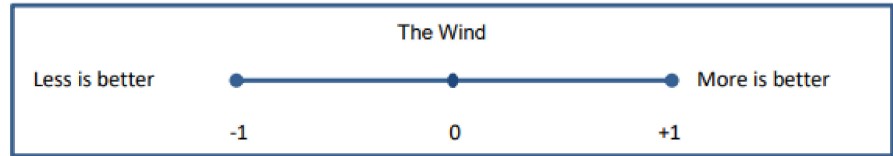

**Figure A1.** The content of questionnaire paper; The TSV, WFSV, and HSV questions are written on the C part.

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
