# Peer review of "A Relationship between Micro-Meteorological and Personal Variables of Outdoor Thermal Comfort: A Case Study in Kitakyushu, Japan"

_sustainability, doi:10.3390/su132413634_

Round 1
Reviewer 1 Report
Enclosed

Author Response
Dear Reviewer 1,
Thank you for your help to improve the paper's quality.
There are some improvements:
1) The term and the title have been changed from 'environmental and human factor' to 'micro-meteorological and personal variables'.
2) The layout has been changed on the introduction part and the material and methods part.
3) On the results part, a new subchapter for the analysis of PET and 3 personal variables (TSV, HSV, and WFSV) has been added.
4) The clothing insulation information has been added to the material and method part.
5) Mistakes and unclear statements have been changed or removed.
We are looking forward to hearing from you.
Warm regards.

Reviewer 2 Report
This is a well-written and well-organized paper that addresses an important topic related to outdoor thermal comfort.
Author Response
Dear Reviewer 2,
Thank you for your help to improve the paper's quality.
There are some improvements:
1) The term and the title have been changed from 'environmental and human factor' to 'micro-meteorological and personal variables'.
2) The layout has been changed on the introduction part and the material and methods part.
3) On the results part, a new subchapter for the analysis of PET and 3 personal variables (TSV, HSV, and WFSV) has been added.
4) The clothing insulation information has been added to the material and method part.
5) Mistakes and unclear statements have been changed or removed.
We are looking forward to hearing from you.
Warm regards.

Reviewer 3 Report
Dear authors,
here below you will find some general observations. Minor details can be found in the pdf marked file.
Best regards.
- The introduction does not provide enough elements of discussion in terms of human factors (please pay attention when this term is used), outdoor thermal comfort assessment, and the peculiarities of the present investigation. You are invited to provide robust references about subjective assessment of thermal comfort outdoor and possible correlation with objective metrics (e.g. PET, UTCI). Some elements about the role of human factors upon environmental quality can be found in: https://doi.org/10.3390/atmos12101272
- The materials and methods section does not contain important info about: i) metabolic rate of the investigated sample; ii) clothing insulation values used in Rayman model; iii) the effect of wind and body movements on clothing thermophysical properties iv) age and sex and other personal details.
- The results and discussion section does not report info about possible gender-related differences. It is not acceptable in a paper of the third Millenary. In addition, figures 11, 13 and 14 are unclear. Particularly it is not clear the meaning of the “PET Standard Predicted”. This is also in figure 15: what is the meaning of HSV Standard predicted?
- Conclusions do not report the limitations of the current investigation. For instance, why the authors did not measure directly tmr to compare their results with simulations provided by Rayman? How reliable can be considered the simulations in the absence of measured data of clothing insulation and metabolic rate? In the perspective of sustainability, why there is not a discussion about gender-related differences in thermal comfort (and other issues) perception?

Author Response
Dear Reviewer 3,
Thank you for your help to improve the paper's quality.
There are some improvements:
1) The term and the title have been changed from 'environmental and human factor' to 'micro-meteorological and personal variables'.
2) The layout has been changed on the introduction part and the material and methods part.
3) On the results part, a new subchapter for the analysis of PET and 3 personal variables (TSV, HSV, and WFSV) has been added.
4) The clothing insulation information has been added to the material and method part.
5) Mistakes and unclear statements have been changed or removed.
We are looking forward to hearing from you.
Warm regards.

Round 2
Reviewer 1 Report
Dear Authors,
my question are enclosed as comments in text.

Author Response
Dear Reviewer 1,
Thank you very much for your guidance and advice.
There are some changes in this new manuscript.
- The term "parameters" in Figure 1 has been changed into "The types of the variable" to prevent terminology misuse in the study since TSV is not the parameter of outdoor thermal comfort.
- The explanation of clothing insulations values and its method of investigation has been added in lines 273-279.
- The explanation of Tmrt values and its method of investigation has been added in lines 286-288 and lines 299-301.
- The explanation of the difference of scale has been added in lines 335-341.
- The values which are written in figures 11, 12, and 13 have been revised.
- The typing mistake in line 541 (now 561) has been revised.
- The inappropriate grammatical writing in line 566 (now 586) has been improved.
- The appendix of the questionnaire (lines 648-657) has been added as a data supplemental to explain the type of scale used in the questionnaire paper.
Thank you for your advice.
Best regards.

Reviewer 3 Report
Dear Authors,
Thank you very much for your efforts.
Best regards.
Author Response
Dear Reviewer 3,
This is the new manuscript (please see the attachment) that has been improved according to the comments of the reviewers. Thank you for your guidance.
Best regards.

Round 3
Reviewer 1 Report
Dear Authors,
I am sorry but in my opinion you shoud more explitielly describe your mrthod of tmrt estimation? Rayman need specific data to meak estimatin.
Kind regards
Author Response
Dear Reviewer 1,
First of all, we appreciated your helpful comments.
We have added the description of the method of Tmrt estimation which is calculated by the Rayman Model software. A detailed explanation is provided in the manuscript from lines 299 to 325 (please see the attachment).
Thank you for your advice and guidance.
Best regards.
